# Last Aid Training Online: Participants’ and Facilitators’ Perceptions from a Mixed-Methods Study in Rural Scotland

**DOI:** 10.3390/healthcare10050918

**Published:** 2022-05-16

**Authors:** Leah Macaden, Kirsten Broadfoot, Clare Carolan, Kevin Muirhead, Siobhan Neylon, Jeremy Keen

**Affiliations:** 1Department of Nursing & Midwifery, University of the Highlands & Islands, Inverness IV3 5SQ, UK; leah.macaden@uhi.ac.uk (L.M.); kevinmuirhead106@gmail.com (K.M.); 2Sterena Consultancy, Cromarty IV11 8XA, UK; sterenaconsultancy@gmail.com or; 3Anschutz Medical Campus, University of Colorado, Aurora, CO 80045, USA; 4Highland Hospice, Inverness IV19 1AF, UK; s.neylon@highlandhospice.org.uk (S.N.); j.keen@highlandhospice.org.uk (J.K.)

**Keywords:** public health, Last Aid, online, death literacy, perceptions, participants, facilitators, mixed methods, Scotland

## Abstract

(1) Background: Palliative and end-of-life care services are increasingly gaining centre stage in health and social care contexts in the UK and globally. Death and dying need are relational processes. Building personal and community capacity along with resilience is vital to support families and communities to normalise death and dying. Last Aid Training (LAT) is one such innovative educational initiative which teaches the general public about the fundamentals of palliative care and promotes public discussion about death and dying. The Highland Hospice [HH] in Scotland has pioneered delivery of LAT in face-to-face settings since March 2019 and online since March 2020 to accommodate pandemic restrictions. (2) Methods: This study used a mixed-methods approach, combining an online survey with LAT participants followed by individual semi-structured qualitative interviews with both LAT participants and facilitators. The primary aim of this study was to investigate the impacts of LAT for participants at the individual, family, and community levels, as well as explore participant and facilitator experiences and perspectives of LAT in an online environment. (3) Results: Overall, this evaluation demonstrates that provision of foundational death literacy education in social contexts enhances the personal knowledge, skills, and confidence of individual community members and supports the notion that this personal growth could lead to strengthened community action. (4) Conclusions: Findings from this study concluded that there is potential to include LAT as the foundational core training to promote death literacy in communities with further exploration to integrate/align LAT with other national/global end-of-life care frameworks.

## 1. Introduction

By 2040, the number of people with increasingly complex palliative care needs in Scotland is projected to rise by 20% [1]. A recent Scottish expert consultation identifies the need to build community capacity and resilience to support informal palliative caregiving so people can remain at home if they so wish [2]. One mechanism to build community capacity is to adopt a public health approach to palliative care as informed by the Ottawa Charter for Health Promotion [3,4]. ‘Compassionate Communities’ uses a community development framework to enhance personal capacity and community empowerment, and is promoted as a public health-promoting palliative care initiative [5]. Recognising palliative and end-of-life care as key issues of major public interest, Scotland’s national framework for palliative and end-of-life care for 2016–2021 is committed to driving a new culture of openness about death, dying, and bereavement [6].

Death literacy has been defined “as a set of knowledge and skills that make it possible to gain access to understand and act upon end-of-life and death care options” and considered an outcome of peoples’ experience of death and dying as well as formal and informal learning [7]. Death education initiatives aimed at the general public to raise citizens’ awareness and knowledge of palliative care to enhance death literacy align well with the theoretical underpinnings of health-promoting palliative care and public health approaches to death and dying [8]. The Kerala model has successfully changed the narratives around death and dying, redefined care at the end of life, influenced national policy, and challenged global models of end-of-life care through community participation and empowerment [9].

Last Aid is an innovative instructor-led educational initiative developed by George Bollig which teaches the general public about the fundamentals of palliative care and promotes public discussion about death and dying [10]. It endorses a simple concept paralleling the universal promotion of First Aid training to enhance citizens’ everyday capacity to deal with minor injury or illness, in that Last Aid can similarly enhance citizens’ everyday capacity to support death and dying in their communities [11]. Since its inception, Last Aid has grown globally and has been supported by the European Association of Palliative Care’s Taskforce on Last Aid and Public Palliative Care Education [7].

Last Aid Training (LAT) is delivered as four short modules over three hours focused on care at the end of life; advanced care planning and decision making; symptom management; and cultural aspects of death and bereavement. The course materials were developed by an international working group [10] and aim to (i) create an awareness of, and discussion on, topics related to death and dying, (ii) build capacity for a more formal palliative care befriending and facilitator role, and (iii) develop a range of Last Aid international age-specific courses particularly for young people [7]. In addition to didactic learning using the course material, informal sharing of experiential learning by course participants is supported by Last Aid facilitators.

Through its development of personal skills and capabilities, Last Aid is a key means of enhancing death literacy and furthering the aims of building community capacity to develop Compassionate Communities, including workplaces. Hospices, in particular, are well-positioned to support such public health care approaches within the communities they serve [12]. In Scotland, the Highland Hospice has pioneered delivery of LAT in face-to-face settings since March 2019 and online since March 2020 via Zoom to accommodate pandemic restrictions. LAT is delivered by a pair of facilitators, drawn from a larger cadre of eight trained facilitators. Facilitators are everyday citizens trained via HH. The shift to online delivery means that participants living outside the region can attend; however, LAT enrolment data confirm that the majority of LAT participants are drawn from the Highland locality (Highland Hospice 2021). A total of thirteen online sessions have been delivered since the beginning of the COVID pandemic.

Whilst recent evidence supports the feasibility and acceptability of LAT both in face-to-face and online environments, the need to evaluate the impact of public health interventions beyond course evaluation has been argued [13,14].

The primary aim of this study was to investigate the impacts of LAT for participants at the individual, family, and community levels, as well as explore participant and facilitator experiences and perspectives of LAT in an online environment. Specific research aims explored participants’ perceptions of their knowledge and confidence on end-of-life care before and after LAT; participants’ experiences on providing end-of-life care before and after LAT; participants’ perspectives on the online delivery of LAT, and facilitators’ experiences of delivering LAT both in online and offline settings.

## 2. Materials and Methods

This study used a mixed-methods approach, combining an online survey with LAT participants followed by individual semi-structured qualitative interviews with both LAT participants and facilitators. Data were collected between October 2020 and April 2021.

### 2.1. Participants

Both LAT participants and facilitators were recruited through HH.

#### 2.1.1. LAT Participants

The publicly available LAT registration form on the HH website was amended to include an option for participants to opt in to participate in this evaluation. In the absence of the registrant’s preference, the Last Aid coordinator from HH sent a follow-up email to clarify participation in the evaluation. If the registrant had opted out of the evaluation, no further contact was made.

LAT participants who indicated their interest in participating received a follow-up email with the Participant Information Leaflet (PIS) and a short electronic flyer on the project with the contact details of the research team. Participants were sent a secure link to the survey, with informed consent obtained electronically prior to survey access. Participants were also able to indicate their preference to take part in a one-to-one interview within the survey.

To recruit interview participants, the Research Associate on the project team worked in collaboration with the Last Aid coordinator from HH. All participants were required to sign the consent form and send the Research Associate a scanned copy via email to then arrange a convenient date and time for the virtual interview via WebEx. The Research Associate established identity and verbal consent prior to proceeding with the virtual interview that was video-recorded.

Alongside written communication, Last Aid facilitators also shared information on the ongoing evaluation at the close of each online training session and invited participants to take part. The Hospice Communications team also posted two general invites via the Hospice’s Facebook and Twitter to aid recruitment of participants to this study.

A total of 105 people opted in to participate in the evaluation at the time of registration. Subsequently, 68 people attended training sessions, with 26 participants completing the survey, giving a response rate of 38%. Participants were predominantly female (*n* = 22) and aged between 35 and 64 years (*n* = 19). Most participants were employed (*n* = 21), with around 40% (*n* = 11) working in health or social care. More than half of the participants were educated to graduate level or above. Reasons for attending LAT were most frequently related to employment or attributed to personal interest. Most participants did not have previous experience of working with people at end of life, but had professional experience with grief and loss and had attended similar training previously. Six participants (two male and four female) completed interviews online for the study.

Demographic information for participants is shown in Table 1. Information on participants’ backgrounds and experience is shown in Table 2.

#### 2.1.2. LAT Facilitators 

LAT facilitators were contacted by the Last Aid coordinator at HH via email to inform them about the study. The email included a participant information sheet and a consent form. A follow-up email was sent two months later to remind facilitators about the on-going study. Participants with interest to participate in the study contacted the Research Associate to then arrange a convenient date and time for the virtual interview via WebEx. The Research Associate established identity and verbal consent prior to proceeding with the virtual interview that was video recorded. Five LAT facilitators (one male and four female) were interviewed in this study.

### 2.2. Data Collection

This study used two forms of data collection. An online survey contained both quantitative and qualitative items and was distributed solely to LAT participants. The semi-structured interviews were conducted using interview topic guides with both LAT facilitators (Appendix A) and participants (Appendix A).

#### 2.2.1. Online Survey

The survey was designed to gather (i) participant knowledge and experiences of Talking about Death and Dying, (ii) Knowledge and Confidence around Death and Dying, and (iii) Decision making and intentions to support end-of-life care needs. In the following section, we present both quantitative and qualitative survey data addressing these topics.

The online survey aimed to evaluate the impact of LAT on participants’ knowledge, understanding, awareness, and confidence related to palliative and end-of-life care issues and decision making and their perceptions of the LAT. The questionnaire (Appendix A) was informed by the Death Literacy Index Questionnaire [15], enabling evaluation of key features of death literacy, i.e., knowledge, skills, experiential learning, and social action as an intended outcome of LAT. The survey also included evaluation of the content and delivery of LAT.

The survey questionnaire was administered using the Jisc Online Survey (JOS) platform. JOS is designed for education and research and the University of the Highlands and Islands holds a license for its use [16].

#### 2.2.2. Interviews

One-to-one semi-structured virtual video interviews were conducted through WebEx. Participant interviews ranged from 28 to 42 min long, with a mean interview time of 35 min. Facilitator interviews ranged from 17 to 40 min long, with a mean interview time of 29 min. All interviews were conducted by the Research Associate and audio from the interviews was extracted to ensure participant anonymity. Data were transcribed verbatim by a professional transcription service.

### 2.3. Data Analysis

Quantitative survey data were analysed using descriptive statistics. The sample size was small (*n* = 26) and not suitable for analysis using inferential statistics/significance testing. Survey data were downloaded from JOS and analysed in Microsoft Excel Version 2013. Participant responses on questionnaire items were captured using a one-time survey from two time periods—pre- and post-LAT. They were asked to provide data for items on the questionnaire pre- and then post-LAT training, where they were reflecting back on their perceptions/experiences before attending LAT rather than having completed the survey at different time points.

Either percentage values of Likert-scale responses or median responses were reported for questionnaire items. Textual survey data were analysed thematically and reported with representative participant quotes.

Qualitative interview data were analysed thematically [17]. For both facilitator and participant samples, one transcript was independently coded by each member of the research team (Kirsten Broadfoot, Leah Macaden, and Clare Carolan). The research team then met to compare and assess emergent coding and the preliminary set of codes and themes derived by the Research Associate. From this, a coherent set of codes and themes was derived and applied to the remaining transcripts.

## 3. Results

This study originally had four specific research aims involving the exploration of participants’ perceptions of their knowledge and confidence on end-of-life care before and after LAT; participants’ experiences on providing end-of-life care before and after LAT; participant perspectives on the online delivery of LAT, and facilitator experiences of delivering LAT both in online and offline settings. Given the mixed-methods study design and presence of quantitative and qualitative data for LAT participants, and solely qualitative data for facilitators, findings are presented to align with combined objectives as follows:Participant perceptions of their knowledge, confidence, and experiences providing end-of-life careParticipant perspectives on the online delivery of LAT and facilitator experiences of delivering LAT in online and offline settings.

### 3.1. Participant Perceptions of Their Knowledge, Confidence, and Experiences Providing End-of-Life Care

The main findings for this combined objective were gathered through the quantitative and qualitative items from the online survey. However, participant interviews also provided rich insights into their experiences of LAT.

#### 3.1.1. Knowledge and Experiences Talking about Death and Dying

Participants indicated high levels of comfort talking about death and dying before attending LAT, with only 15% of participants responding that they would avoid the topic of death and dying, and 4% indicating they would avoid people who were grieving. After LAT, participant discomfort with these activities fell to zero. Prior to LAT, when asked why they might avoid conversations on death and dying, 27% of participants indicated they might avoid conversations due to fear of upsetting others, and 19% due to personal discomfort. After LAT, none of the participants indicated they would avoid conversations due to personal discomfort and only 4% indicated they would avoid conversations due to fear of upsetting others (Figure 1).

When asked where and when they discussed death and dying, participants reported a 20% increase in discussing death and dying with family members, a 19% increase within their community, and 11% in the workplace after LAT. No increase in discussions within church or religious settings was reported. Notably, the religious setting was not applicable to 65% of respondents (Table 3).

Participants also reported relatively high levels of confident conversations on death and dying prior to participating in LAT, with 85% of participants indicating they were at least somewhat confident discussing death and dying with a close friend, 62% with a child, 65% with a recently bereaved person, 54% with a GP about support for a dying person, and 54% were confident when talking to someone who is dying. After LAT, these percentages increased to 100%, 88%, 96%, 100%, and 96%, respectively, as seen in Figure 2.

#### 3.1.2. Knowledge and Confidence around Death and Dying

Beyond participant comfort and confidence in holding conversations on death and dying, participants were also asked to rate their knowledge of end-of-life care and community support available. Figure 3 presents median response values demonstrating an increased value for the majority of items following LAT.

Participants were then asked to rate their confidence in providing care and support to people who are dying. As seen in Figure 4, confidence levels increased in all domains (administering medication, moving and handling, bathing, assisting with eating and drinking, using non-pharmacological intervention to increase comfort, and symptom recognition) following LAT.

### 3.2. Decisions and Intentions to Support End-of-Life Care Needs

In the final section of the survey, participants were asked to indicate whether they had undertaken or intended to undertake various measures relating to end of life pre- and post-LAT. There was an increase in the number of participants who indicated they had undertaken or intended to undertake all the measures after the training. The greatest increases (>30%) were observed in practical measures involving future planning (e.g., making a will, advance care planning, and financial planning) and practical aspects of providing palliative care, as seen in Figure 5.

In summary, data from the survey clearly indicate that LAT had an impact on participant preparedness, comfort, and confidence in discussing death and dying, as well as their decisions and intentions to support end-of-life care needs. Open-ended survey items and participant interviews were also used to gather more in-depth insights of participant experiences of LAT. Themes and data from these qualitative sources are presented in the next section.

### 3.3. Overall Impact of LAT on Participants

Thematic analysis of both participants’ free-text survey responses and interview data produced three main themes: (i) demystifying palliative care and enhancing understanding, (ii) creating normalised conversations around death and dying, and (iii) thinking ahead and advocating for self and others.

#### 3.3.1. Demystifying Palliative Care and Enhancing Understanding

As indicated in the survey data, participants’ existing understandings of palliative or end-of-life care were split along personal experience and professional expertise. Participants who were professionally expert in the area described little change in their knowledge from attending LAT. However, participants with no prior understanding described changed sensemaking of personal experiences of dying and changed understandings of palliative care. The latter included differentiation between palliative and end-of-life care, care beyond professionally delivered care, and alleviation of suffering as a primary goal.


*“I don’t know about not understanding about it but from a personal, life … somebody who was ill in my own family and obviously I didn’t know what to expect then, but by the end of this training I learnt much more about it and it made me understand what I went through … and if this ever happened again, I’d have a greater understanding, it helped me deal with it”.*
(Interview Participant 3)

Participants also described enhanced knowledge about access to a larger support network and voiced a better understanding of stages and structured actions related to death and dying (reduced death anxiety, clarity of legal aspects of death and dying, anticipatory care planning, and grief and grieving) and individualised responses to death and dying within society.


*“Well, the main thing was that I learnt more (…) various stages, how you should not be frightened of it. And it’s…just a good (…) thing for you to learn, you know more about what may happen and also afterwards, like the grieving bit, there’s no right or wrong way that people grieve”.*
(Interview Participant 3)

#### 3.3.2. Creating Normalised Conversations around Death and Dying

When asked about applying LAT to everyday life, participants reported a need to find conversational openings to normalise death and to tailor conversations across difference, highlighting the amount of awareness and confidence raised through their own clearer understanding of death and dying processes and how to serve those who are dying. Overwhelmingly, participants reported the need to talk more about death and dying as a normal part of life.


*“How positive it is for society to have accepted that it’s a good, kind, loving thing to be able to talk about death”.*
[Survey respondent 4]

All participants shared that LAT had boosted their confidence in being able to have constructive conversations, especially difficult family conversations around death due to gendered emotional expressions and fears of negatively impacting interpersonal family dynamics.


*“I already know and already talk about these things but as I say, I think the course did make me think ‘yeah, we need to talk about them more frequently and not when it’s about to happen’, you know, you need to talk about them just over Sunday dinner when nobody is dying. That’s probably the key thing; let’s not wait until it’s needed, let’s talk about it all the time.”*
(Interview Participant 11)

However, the impact of COVID restrictions on meeting people outside their friends and families had placed limits on opportunities for such conversations.


*“I would say because of the current situation again I have not really had much opportunity to. Not a matter or bias to the course, I mean, it’s only been a matter of weeks and we’ve not had the opportunity to put any of these choices into practice”*
(Interview Participant 10)

#### 3.3.3. Thinking Ahead and Advocating for Self and Others

Finally, pro-actively planning for one’s own future and encountering death was a significant outcome of LAT. Participants described getting their own ‘house in order’ including conversations to be had, and practical planning such as legal and funeral arrangements.


*“I don’t have power of attorney and that’s the thing that we talked about quite a lot. So yes, it’s made me think very seriously about that and in fact my husband and I have been speaking about that. I teach it but I don’t do what I preach.”*
(Interview Participant 5)

Moreover, ‘thinking ahead’ impacts described above extended beyond their own individual perspective to encompass those within their family networks. With increased confidence in conversations around death and dying, participants felt a need to advocate for such conversations within their families and wider social networks.


*“I am a great advocate of conversations and I know that they don’t happen, people don’t want to talk about death or dying, I am quite an advocate that people need to do that. So, it almost gave me permission to carry on doing that.”*
(Interview Participant 5)

Participants also reported feeling better equipped to guide close or intimate others as they had a better understanding now of their own experiences and could be of better support to those in need. LAT enabled increased awareness and confidence in participant preparations and planning for the future including seeking permission to advocate and discussions around legal processes, wills, future wishes, preferred place of care, etc.


*“There were certain things that we talked about that were more vague ideas that are now relatively clear as far as knowing what would need to be done. And in terms of my own life, it’s put into focus that my wife and I should start looking at things like getting wills and things like that, regardless of the fact that we’re only in our mid-thirties, it’s never too early.”*
(Interview Participant 9)

In summary, participants experienced not only increased knowledge and understanding of palliative and end-of-life care, but also felt more confident to initiate and hold conversations as well as prepare, plan, and advocate for self and others at end of life.

### 3.4. Participants’ and Facilitators’ Perceptions and Experiences of LAT within the Virtual Online Environment

As shown in the findings from the first combined objective, there is no doubt that LAT had a significant impact on participants. However, there was some uncertainty as to the effect a move to online delivery might have on the experiences of both participants and facilitators. Objectives 3 and 4, combined here to represent the perceptions and experiences of participants and facilitators, sought out such data through semi-structured interviews predominantly, with some additional insights gleaned from the online survey.

It should be noted that, unlike participants surveyed and interviewed for the study, LAT facilitators had experience of both online and in-person session delivery. To address this difference, while both participant populations were asked a series of questions on course design, content and delivery, facilitators were also asked to compare experiences. Thematic analysis of all interviews revealed four analytical themes: (i) accessibility, (ii) desired diversification, (iii) connectedness, and (iv) discomfort and difficulties.

#### 3.4.1. Accessibility

The convenience and flexibility of virtual online delivery was a widely reported strength of LAT, enhancing access to LAT and mitigating geographical constraints of delivering LAT in remote and rural contexts.


*“We can’t physically manage to take presenters and take facilitators out to small rural communities in the Highlands. Highlands geographically—it’s so difficult. And there is a lot to learn, I suspect, from doing it on Zoom—doing it online—in terms of taking it out to people in little remote rural communities, where they don’t have access to anything. I do think that online sessions in all sorts of areas have a huge potential.”*
(Interview Participant 7)

While participant survey data indicated that no modality of course delivery was preferred over another, online LAT was reported as offering flexibility in scheduling and accessibility to those with diverse abilities and circumstances. Some participants reported online delivery facilitated easier access to LAT material, which was perceived as beneficial.

*It was a good mix of being interactive with trainers and other participants and engaging solo through listening or reading. The trainers were very approachable and relatable*.[Survey respondent 5]

*Ability to attend! I probably wouldn’t have found the time otherwise. Home comfort. Being able to take notes without feeling rude or not present*.[Survey respondent 6]

Although most participants did not report limitations with online delivery of LAT, there was a general consensus from participants and facilitators about potential threats to access including digital connectivity and the need for technical support, as well as a call for LAT to be able to accommodate diverse levels of digital literacy.


*“We’ve got to learn how to use them and to learn how to get the best out of them—and it’s a steep learning curve and an uphill struggle.”*
(Interview Participant 7)

#### 3.4.2. Desired Diversification

When asked about course content, participants enjoyed the content and spoke clearly to its flow, format, and modular design. However, one in five survey participants indicated that they would welcome additional content, such as integrating philosophical perspectives and examples from real human experience and non-traditional relationships.

*Content could do more to recognise secular philosophical aspects of death/dying & life/living, as well as addressing issues surrounding death/dying and caring for someone who is dying in circumstances where partnership is non-traditional, i.e., LGBTQIA+ relationships*.[Survey respondent 7]

Similarly, in interviews, some participants voiced a need for topic diversity and illustrative examples, including discussions of different types of death and how to prepare for them, alternative arrangements, or contingencies to be aware of and how to manage unexpected death.


*“something I think could be improved, in the way they want to take it out to general communities and things, was it was very, very heavily about—the context of the death that they were talking about was very much a predicted, expected planned death and I guess there’s room for that course to be much wider, to include people having at least some thoughts and discussions about what if somebody doesn’t die in a predicted or expected way.”*
(Interview Participant 11)

Likewise, LAT facilitators expressed the desire to have some flexibility and the ability to use one’s own notes or scripts to deliver the course rather than deliver from the prescriptive guidance notes made available to them. 


*“I think a script. Although now, it’s funny … the more I think about that now, actually I wouldn’t like that because I’ve tried using other people’s notes and I just re-write them because they are not my words, they are not how I would say it. So there was a handbook and it was being translated but actually I would still want my own notes because that’s not how I would say it, it’s not me, so it doesn’t sit comfortably for me to say it that way.”*
(Interview Participant 6)

#### 3.4.3. Connectedness 

While participants described LAT as informative, person-centred, meaningful learning delivered in a sensitive and collaborative way, connectedness or a sense of connection between participants and facilitators was critical to facilitating learning and enhancing the experience.


*“If I were speaking to somebody who was about to take the training and didn’t know what to expect I’d say, ‘don’t worry, you are not going to leave curled up in a ball in fear of the unknown’, it’s … you know, obviously it’s a topic that’s difficult for a lot of people to discuss but I feel that it was done in such a way it was as sensitive to that as it could have been and it was, the information that was provided it helped to disarm the topic a bit. It made the subject a little bit less scary, I guess.”*
(Interview Participant 9)

Facilitators described the willingness of people to share their stories and were glad to be able to provide help and connect to others who were enduring loss. However, the fragile and temporally constrained nature of connectedness in online delivery was also evident.


*“Just so people are willing to share this very private part of their lives and to open up the conversation about dying … it’s part of a bigger, national movement to talk about death and dying as a normal part of life. And sometimes I’d like to be just a little fly on the wall in the corner of the room just looking down on that conversation because you just don’t appreciate sometimes how profound that is, I think we take it for granted that these people are opening up their lives in the little TV screen in their living room for that short time. And then they are gone, they click a button, and they are gone, and you have no more connection with that person ever again, but you just had that little opportunity to just drop something positive into their experience and hopefully help them along the way.”*
(Interview Participant 1)

As such, connectedness was often challenged by a lack of physical proximity, group composition, and tailored connections.


*(a) Physical proximity*


Online delivery of LAT was perceived to limit formal and informal opportunities to interact with other participants and inhibited opportunities for group discussion and spontaneous and natural conversations. Participants missed physical human connection and the ability to pick up on the body language of others and provide emotional support if necessary.


*“Subject is difficult and emotional—delivering the course online removes the element of human contact and support that some may find helpful in dealing with these matters, even in an educational setting”.*
[Survey respondent 7]

Similarly, LAT facilitators compared their experiences of online delivery to face-to-face/in-person delivery and stated that online delivery was more impersonal, describing difficulties with identifying and discussing emotions. Facilitators felt that there was greater ability to hold emotive discussions, use more anecdotes, and hold more informal conversations during face-to-face delivery, allowing for more intuitive connections and a better participant experience.


*“… you can have these really honest conversations when you are face-to-face and you can say to them, if you need a minute you can go out and you can read people’s emotions”.*
(Interview Participant 2)


*“In a room, you can go round afterwards and say, ‘Do you want to look at the pieces of paper?’ and chat to people. Or pick out someone who’s been uncomfortable, or the quiet one in the corner who’s said very little. You’ve got that ability to relate to people that you just don’t have on screen.”*
(Interview Participant 7)

Some participants who had only completed LAT online imagined that being face-to-face would allow the sessions to be more interactive, provide more support for emotional conversations in small groups, and potentially enhance spaces for discussion. One participant commented on the need to increase spaces for practice in LAT overall so participants could have small role play or practice conversations in small groups or pairs to put the content into application.


*“an improvement to the course, I think, would be to actually get people to have that conversation perhaps in pairs, we did go into little break-out rooms, as can often be the case, people talked about other things and kind of avoided talking about the thing that wanted to be talked about … make it longer and enable people to really have more interaction and more discussion between each other and between themselves and kind of test out having some of these conversations”*
(Interview Participant 11)


*(b) Group composition*


Group composition also had a significant impact on connectedness. Some participants stated that when there was symmetry between facilitators, the course had a natural flow and interaction. However, for participants, diverse or mixed group participants (e.g., not all clinical or professional participants) was seen to promote interaction and connection. Importantly, participants stated that facilitators must not assume homogeneity within groups and needed to appreciate potential audience diversity when discussing issues of faith and cultural practices.


*“it would have been quite nice to have been with other people that were not clinical. That’s the only…I know you just can’t control that and, as I say, I know a few people couldn’t link in that day so there may have been people within that but personally, for me, that would have been good to hear from their viewpoint as well”*
(Interview Participant 5)


*(c) Tailoring connections*


Relatedly, participants shared that gathering information on learners prior to attending the session would enable content to be tailored more to their needs. Such tailoring would ensure course content would be more contextually appropriate. Participants also desired LAT handouts ahead of time so they could be more prepared, enabling them to balance attending to interaction and didactic delivery within the sessions. This preparation would create common ground and allow sessions to be more interactive.


*“And again it kind of begs that question to me, that I’ve obviously got a bit stuck on, is that because they are copyrighted in some way, I mean why on earth would you not give people the slides if they are on a course? It just seems completely wrong to me.”*
(Interview Participant 11)

Overall, then, the dimension of connectedness, so central to the LAT experiences of both participants and facilitators, was dependent on several factors for its accomplishment, some of which could be easily strengthened to improve delivery in an online setting. The final theme in this online experience highlights the dialectical tension between emotionally charged content and human experiences and course delivery modalities.

#### 3.4.4. Discomfort and Difficulties

Whilst participants and facilitators described their experience of LAT as a meaningful learning experience, and online delivery as enabling greater access and flexibility, it was not without participant discomfort or difficulty.

Given the emotional nature of the content, online delivery afforded participants the comfort of learning in their own home environment and enabled them to process emotion privately by turning off sound and camera if so desired. Some participants also stated that online LAT enabled them to learn at their own pace. One participant went further, suggesting that the training could take place over a few days so that the content did not overwhelm the lay audience. Other participants highlighted the physical discomfort of sitting for prolonged periods during virtual training and recommended more breaks.

While participants spoke to the discomfort associated with session design and the volume/nature of the content, LAT facilitators voiced considerable difficulties and discomfort delivering Last Aid due to (a) knowledge base, (b) role autonomy, and (c) online facilitation skills.


*(a) Knowledge base*


Some facilitators expressed concern initially about their perceived lack of knowledge; however, facilitators also shared that being prepared and being honest when you do not know something were important skills. Accessing other resources such as books and papers to reference or address knowledge gaps was perceived to be more challenging in the online environment. Overall, however, facilitators shared that such knowledge gaps were also learning opportunities and made their facilitation experiences meaningful in their own daily lives.

*“I feel that it’s given me a bit more confidence and knowledge behind it and it’s certainly facilitated conversations in my own life”*.(Interview Participant 2)

One of the ways suggested by facilitators to address variability in knowledge was for there to be a foundational script or slide set to be used in their facilitator training and/or the creation of a facilitation guide or booklet to which they could refer. This was seen as increasingly important when there were considerable gaps in time between their training and facilitation as well as session facilitations to mitigate concerns.


*“I think my expectations were that we would probably get a script that we were going to use when we were actually facilitating the sessions. Especially when I don’t have a clinical background and I don’t have a knowledge of end-of-life care in the same way: I see it as a bystander. So I did think we would get a training manual … Definitely a script that would help”.*
(Interview Participant 6)


*(b) Role Autonomy*


Beyond knowledge base, some facilitators described differing degrees of comfort and agreement with course content, experiencing challenges to their autonomy and authenticity as facilitators. Some facilitators struggled with their role and content control as facilitator versus that of the creator. Some experienced this struggle as feeling like an imposter as it was not their content and also being unsure about how individual changes or additions would either augment or distract from the original message and vision of LAT.


*(c) Online facilitation skills*


Finally, technology and the online learning environment itself posed some difficulties for facilitators, as they often felt ‘powerless’, ‘abandoned to the technology’, and the experience as ‘nerve wracking’. Struggles around managing technology including breakout rooms and technological mishaps, navigating time constraints, varying degrees of comfort in clarifying the group’s composition, individual needs, and agendas as well as challenges around public speaking were all shared by LAT facilitators. Some stated they felt more self-conscious within the online environment, feeling a loss of authenticity and professionalism.


*“I feel self-conscious with the technology, I feel it takes away a little bit of my personality, my ability to be my real self because I’m self-conscious. So I think probably in-person but I do see the need for the technology”.*
(Interview Participant 1)

As a result, facilitators wanted more face-to-face discussions, chances to practice co-facilitation before a session, more regular session delivery, and the provision of a manual or booklet for facilitation guidance. They also suggested increased discussion of content and delivery as a group so they could learn from each other and also how to make the most of the online experience.

*“I just need to keep delivering the material regularly so I can keep up my skills. One of my worries is if they take on board too many new facilitators then we all end up just doing one course a year or something. That will just be pointless to me, I think to keep your skills up and your … you tuned in to it, I think you have to keep doing it regularly”*.(Interview Participant 1)

In summary, delivering LAT online posed many of the challenges experienced in online education in general around technological familiarity, connectivity, and the constant negotiations of time and space for authentic interaction. Participants and facilitators expressed that this modality held great promise in terms of granting access and awareness to a wider audience, and the expansion of public understanding of death, dying, and end-of-life care at a foundational level. With some adjustments, participants and facilitators believed that LAT could capitalise on its many strengths and overcome barriers to the construction of a vibrant socio-emotional space for LAT content.

## 4. Discussion

The first aim of this study was to evaluate the impact of the LAT program delivered online by HH. Findings from this evaluation support the utility of Last Aid as an educational initiative to enhance death literacy. Survey and interview data report increased personal comfort in talking about death and dying and engagement with family and wider community networks following LAT. Importantly, LAT encourages citizens to plan for death and to adopt a socially inclusive approach, i.e., encompassing positive practical considerations for their own death but, importantly, advocacy for others within their own social networks. These findings endorse the view that anticipatory care planning should adopt a wide upstream health-promoting approach driven by community awareness raising [18], rather than current approaches embedded in the last few months of clinical care focussing on harm reduction from unwanted treatment. High levels of understanding of Do Not Attempt Cardiopulmonary Resuscitation DNACPR before undertaking Last Aid supports findings from other studies [18]. The latter likely strengthened by increased public debate about DNACPR consequent on the COVID-19 pandemic [19]. Public health approaches such as LAT as tools and vehicles to promote death literacy warrant wider implementation and evaluation across death systems to rebalance death and dying [9] involving families and communities.

Beyond care planning, care-related knowledge and intent to support care (including administration of medicines) had increased following LAT. While survey data post-LAT demonstrated an overall positive trend in confidence in providing care and support at the end of life, only 50% of participants rated themselves as either confident or very confident in doing so. Notably, the proportion of respondents suggesting modification of the ‘relieving suffering’ module was broadly similar to the other three modules, with no new additional content specifically requested. However, data from the interviews emphasised that participants had limited time or opportunity because of restrictions imposed by the pandemic to apply their learning in real-life contexts. This suggests that the theoretical framework of a carers’ ability ‘to know’, ‘to be’, and ‘to do’ [20] had only been partially addressed, in that activation of carer ‘to do,’ in terms of preparedness for task-related care, had not been achieved. Unpicking whether this is simply due to lack of opportunity or whether LAT is sufficient in developing confidence and preparedness in task-related care is unknown and merits further evaluation. Moreover, while findings support increased understanding of how to support others and knowledge of support available within their local communities, limited new interest in facilitating and/or setting up an end-of-life care group within their local communities was expressed.

Overall, this evaluation demonstrates that provision of foundational death literacy education in social contexts enhances the personal knowledge, skills, and confidence of individual community members, and supports the notion that this personal growth could lead to strengthened community action. Strengthened community action was evidenced by normalisation of death and dying together with promotion of anticipatory care planning. Thus, Last Aid has demonstrable outcomes similar to other public health interventions in palliative care in terms of community engagement [21]. Community engagement is different to community development. Whilst community engagement involves processes by which communities and services work together to enhance death literacy, community development is operationalised [22] in delineated changes and outcomes for individuals and collective communities to build capacity for practical support and social care [7]. Within this study, evidence of development of community capacity to accomplish elements of practical care and support at end-of-life was less clear. Moreover, self-sustaining community development of compassionate communities was nascent, with little expressed appetite to support community-derived support networks. Hence, whilst Last Aid achieves demonstrable impact in terms of community engagement, promotion of community development is less certain, with the COVID-19 pandemic identified as a possible barrier to strengthening community capacity at the time of writing.

The second aim of the study was to explore participant and facilitator perceptions and experiences of the online delivery of LAT. The shift to online learning necessitated by the pandemic paradoxically presented both threats and opportunities to participants and facilitators alike. Online delivery offered an accessible and convenient mode of delivery, enabling greater reach for a variety of populations; this includes those with carer responsibilities or those living in remote and rural populations in Highland. However, barriers to digital inclusion were evident, with concerns about connectivity and digital literacy portrayed. Given that evaluation only pertained to participants who had accessed LAT online, the influence of digital poverty did not feature. Finally, duality in the emotional qualities of learning environments was evident; some believed that face-to-face learning promoted greater emotional engagement with others, whereas online learning enabled a safe space to privately process one’s own emotions. Taken together, these findings underscore the salience of social and emotional context of learning beyond the physical and technological context and acknowledge a wide range of learner preference, implying that, beyond the pandemic, delivery of Last Aid should accommodate participant learning preferences and support.

High rates of satisfaction with LAT were expressed both by participants and facilitators. While the content of the four modules was deemed appropriate by most participants, inclusion of culturally competent learning materials to reflect audience diversity, diverse and person-centred perspectives on death and dying such as sudden death, social interaction, experiential learning with equity of learning experience, and access to materials ahead of sessions to ensure participants share common ground at the outset to enable interactive learning, were recommended. Finally, appreciating learners’ backgrounds prior to attendance to enable tailoring or purposefully diversifying groups of learners were proposed as possible mechanisms to enhance inclusive learning.

Facilitation of LAT was highly rated by participants, with LAT facilitators perceived as professional and relatable. Facilitators themselves expressed a need for further facilitator training and support to address unmet learning needs, such as facilitation skills training, and the development of tangible resources such as a facilitation guide. Digital skills development and provision of additional technical support for online delivery was also desired. The need for active regular facilitation practice was perceived as vital for skills development and that development of a community of practice for regular debrief and discussion would further enhance confidence and competence in facilitation skills whilst providing welcome peer support. Support was also conceived as securing ‘permission’ to tailor materials to embed cultural sensitivities and learner inclusivity.

There were several limitations to this study. The sample size for the survey was relatively small. Whilst recruitment issues in palliative care research are recognised [23], additional issues could have impacted on recruitment strategies. First, survey participants might have been overwhelmed with having to complete the short evaluation as part of the LAT which is mandated through licence of the course. Second, survey participants were perhaps unaware of the distinction and purpose between the two evaluations. Finally, the possibility of ‘digital fatigue’ from having to engage with virtual environments on a day-to-day basis during a pandemic might have influenced decision making to participate in the study.

The study was also conducted with one course provider serving an area with a predominately white ethnic population, with the majority of survey participants highly educated, married, middle-aged women with previous experience of grief and bereavement training, suggesting possible risks of selection bias. The evaluation also measured impacts at one time point only post-course completion, meaning that translation of learning in terms of intended impacts versus actual impacts cannot be inferred. Thus, longitudinal evaluation is merited and would provide additional insight into knowledge and skills decay over time.

## 5. Conclusions

To our knowledge, this is the first UK evaluation of Last Aid that provides insights into the outcomes of LAT, thus addressing limitations of earlier evaluations [13,14]. Moreover, this is the first evaluation to include information-rich thick data both with participants and facilitators of LAT delivered online.

Findings from this study concluded that there is potential to include LAT as the foundational core training to promote death literacy in communities, with further exploration to integrate/align LAT with other national/global end-of-life care frameworks.

## Figures and Tables

**Figure 1 healthcare-10-00918-f001:**
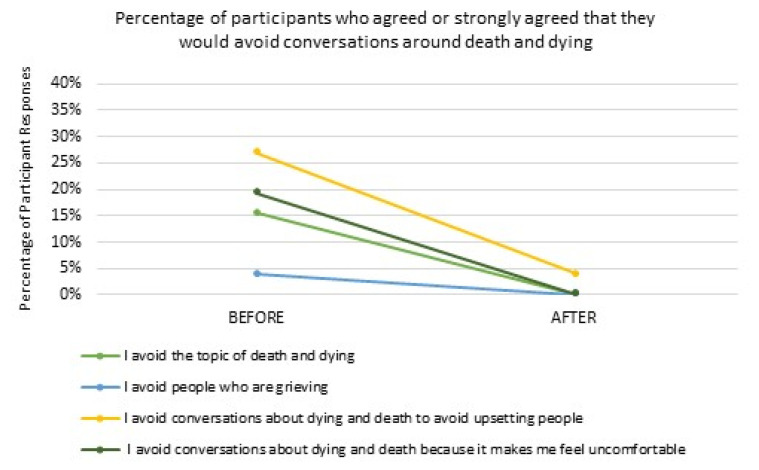
Talking about Death and Dying.

**Figure 2 healthcare-10-00918-f002:**
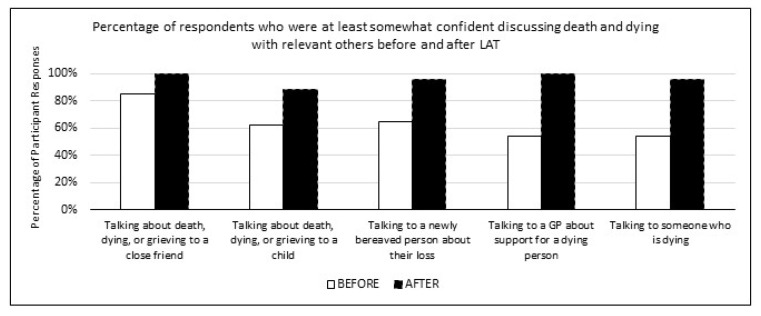
Confident Conversations on Death and Dying.

**Figure 3 healthcare-10-00918-f003:**
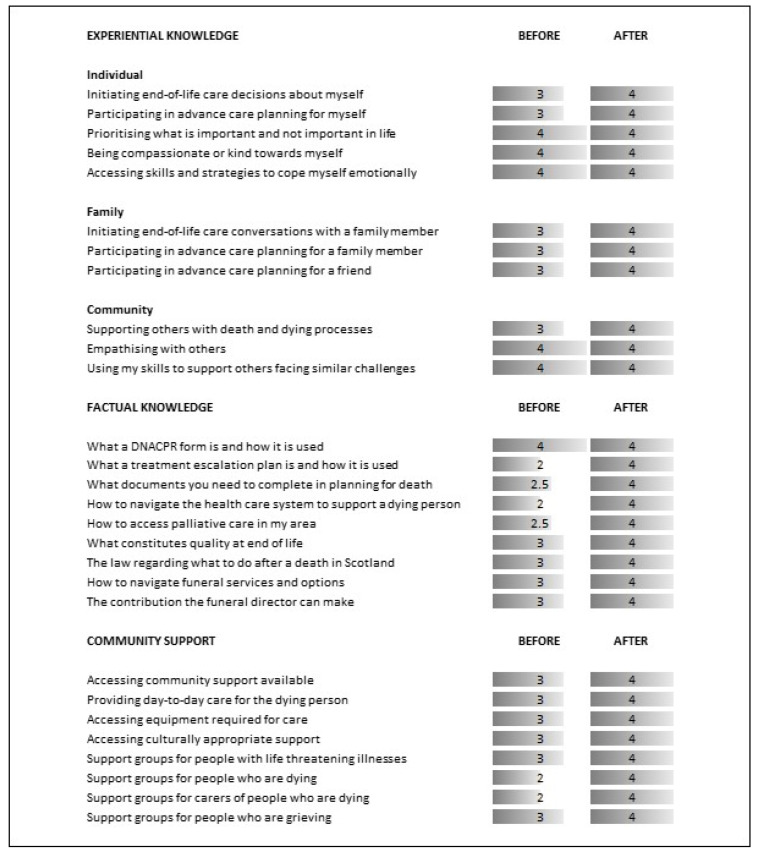
Experiential Knowledge, Factual Knowledge, and Knowledge of Community Support. Experiential knowledge (1 = Not at all comfortable; 2 = Not very comfortable; 3 = Somewhat comfortable; 4 = comfortable; 5 = Very comfortable). Factual knowledge and Knowledge on Community Resources (1 = None; 2 = Weak; 3 = Fair; 4 = Good; 5 = Very Good).

**Figure 4 healthcare-10-00918-f004:**
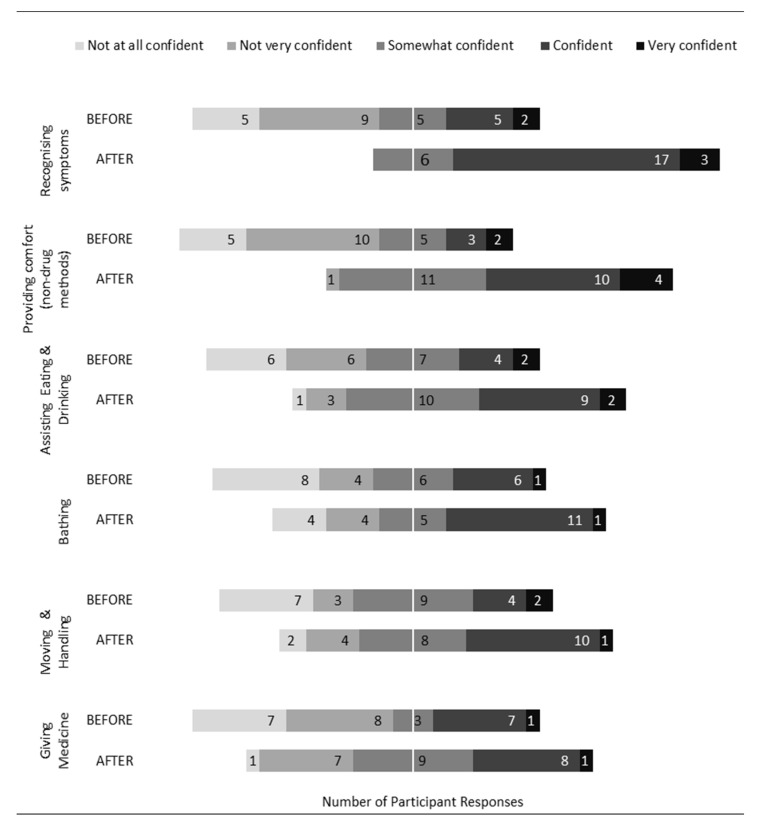
Confidence in supporting care of people dying.

**Figure 5 healthcare-10-00918-f005:**
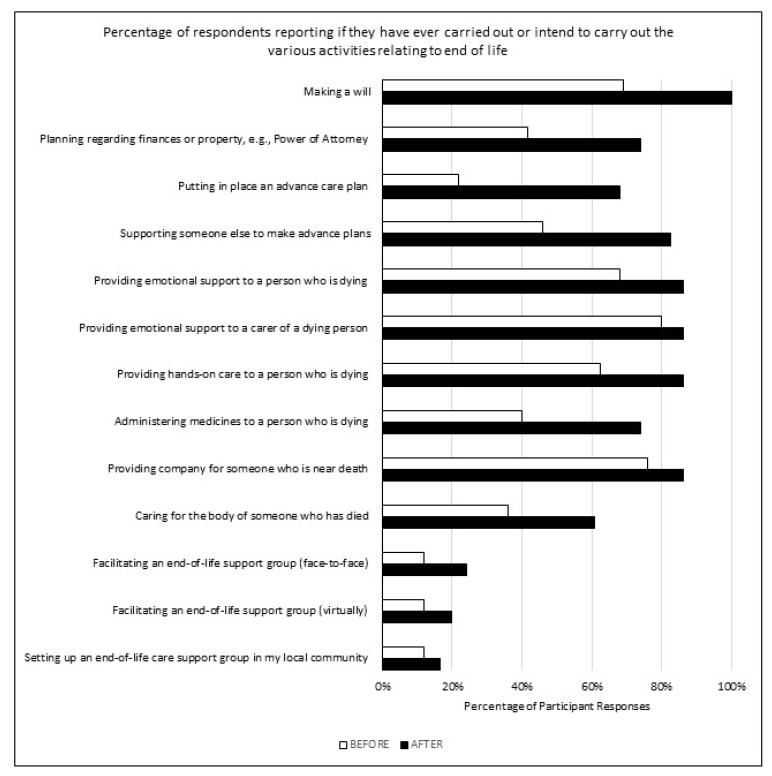
Decisions and Intentions to Support End-of-Life Care Needs.

**Table 1 healthcare-10-00918-t001:** Demographic Information.

**Age**	**N**	**%**	**Gender**	**N**	**%**
18–24 years	1	3.8	Male	3	11.5
25–34 years	2	7.7	Female	22	84.6
35–44 years	4	15.4	Rather not say	1	3.8
45–54 years	9	34.6			
55–64 years	6	23.1			
65–79 years	4	15.4			
**Education**	**N**	**%**	**Employment**	**N**	**%**
National 5 or equivalent	1	3.8	Employed—full-time	10	38.5
Highers or equivalent	2	7.7	Employed—part-time	9	34.6
Certificate or Diploma	5	19.2	Self-employed	2	7.7
Undergraduate degree	8	30.8	Retired	4	15.4
Postgraduate degree	7	26.9	Student	1	3.8
Rather not say	3	11.5			
**Employer**	**N**	**%**	**Residence**	**N**	**%**
NHS	6	23.1	Urban area	10	38.5
Social Care	1	3.8	Semi-Urban area	4	15.4
Third Sector	4	15.4	Rural area	5	19.2
Education	7	26.9	Remote area	2	7.7
Other	8	30.8	Semi-Rural area	5	19.2
**Relationship Status**	**N**	**%**	**Networks**	**N**	**%**
Married	15	57.7	Family	22	84.6
Never married	2	7.7	Neighbours	9	34.6
Widowed	2	7.7	Community Groups	7	26.9
Divorced	1	3.8	Religious Groups	5	19.2
Separated (not divorced)	1	3.8	Friends	22	84.6
Partnered (not living together)	1	3.8			
Single	2	7.7			
Other	2	7.7			
**Reason for Attendance**	**N**	**%**	**Source of Access**	**N**	**%**
Personal interest	21	80.8	Family	1	3.8
COVID-19	1	3.8	Colleague	10	38.5
Long-term condition	1	3.8	Email	3	11.5
Terminal illness	1	3.8	Social Media	4	15.4
Related to work	16	61.5	Hospice Newsletter/Website	11	42.3

**Table 2 healthcare-10-00918-t002:** Background and Experience.

Background and Experience		N	%
Religious/Spiritual Background	Yes	13	50.0
	No	12	46.2
	Rather not say	1	3.8
Religious/Spiritual Practice	Yes	10	38.5
	No	15	57.7
	Rather not say	1	3.8
Experience with people at end of life—Paid	Yes	10	38.5
	No	16	61.5
Experience with people at end of life—Volunteer	Yes	10	38.5
	No	16	61.5
Work experience with grief and loss	Yes	13	50.0
	No	13	50.0
Volunteer experience with grief and loss	Yes	12	46.2
	No	14	53.8
Previous training on dying, grief, or bereavement	Yes	15	57.7
	No	11	42.3

**Table 3 healthcare-10-00918-t003:** Talking about death and dying with others.

Talking about Death & Dying	**Time Point**	**Agree** **(%)**	**Disagree (%)**	**Unsure** **(%)**	**NA** **(%)**
We discuss death and dying in my family	1	57.8	38.4	0.0	3.8
	2	77.0	15.4	3.8	3.8
We discuss death and dying in my community	1	30.8	53.8	15.4	0.0
	2	50.0	38.5	11.5	0.0
We discuss death and dying in my workplace	1	46.2	34.6	3.8	15.4
	2	57.7	26.9	3.8	11.5
We discuss death and dying in my church/ religious gathering	1	34.6	0.0	0.0	65.4
	2	34.6	0.0	0.0	65.4

Timepoint 1 is before LAT and timepoint 2 is after LAT. Agree is the total percentage of participants who either agreed or strongly agreed with the question. Disagree is the total percentage of participants who either disagreed or strongly disagreed with the question. N (Number of Participants); NA (Not applicable).

## Data Availability

Data for this research are stored on the University’s secure SharePoint provided through its Microsoft 365 tenancy. Access to the tenancy, any areas within it, and to the UHI SharePoint implementation is regulated by Active Directory and by specific allocation of users to permissions groups for restricted access areas. Access to anyone not employed by the University is only permitted after a request to create an external account for access to specific areas is approved. These accounts have restricted access and duration regulated by the account requester as well as centrally by the University Archivist and Records Manager and designated staff in the IT department. On conclusion of the research, the University Archivist and Records Manager must be consulted for instructions on how and when to dispose of data in line with the University Partnership Retention and Disposal policy.

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
