# Peer review of "Last Aid Training Online: Participants’ and Facilitators’ Perceptions from a Mixed-Methods Study in Rural Scotland"

_healthcare, 2022, doi:10.3390/healthcare10050918_

Round 1
Reviewer 1 Report
This study provides a valuable contribution to palliative care research and considers the wider impact that end of life has on those who experience it. The study demonstrates the benefits in the LAT as a form of support, in particular for those who have no previous experience. The article contains some qualitative results which appear to be inconsistently referenced. (Either survey respondent or P. xxx) if this is intentional is there a way of explaining what this refers to? Alternatively, do these references almost become identifiers and therefore need to be anonymised to a higher level?Author Response
The authors would like to thank the Reviewer for the helpful suggestions and comments.
|
Reviewer 1 |
Responses |
|
The article contains some qualitative results which appear to be inconsistently referenced. (Either survey respondent or P. xxx) if this is intentional is there a way of explaining what this refers to? Alternatively, do these references almost become identifiers and therefore need to be anonymised to a higher level? |
These responses have now been identified as Survey Respondent 1, 2, 3 etc. |
|
Page 2/Line 52: What are the specifics of the Kerala model? Could it be explained briefly, or is it described in reference 9?
|
We believe that description of the Kerala model would further lengthen this already long manuscript. Hence, we have elected not to describe in detail here. However, we believe that the reference cited gives sufficient insight into the model is so desired by readers.
|
Reviewer 2 Report
General comments:
This study aims to evaluate a new educational initiative named “Last Aid Training” in terms of the impacts on participants who took the course and explore how facilitators (giving the course) and participants (taking the course) experienced the delivery online. The course is directed at the public (non-professionals). Results of the evaluation seem promising because, for example, personal knowledge of death and dying was enhanced. The authors conclude that there is potential to include the course in foundational core training to promote death literacy in communities.
The initiative to develop such a course and spread its use is highly relevant and motivated by the increasing share of people who will need palliative care in the future. Both formal and informal learning is essential in informing the public about palliative care and end-of-life care.
Thanks for a well written original article.
I only have some detailed comments concerning the methods and results sections.
Specific comments:
Introduction
Page 2/Line 52: What are the specifics of the Kerala model? Could it be explained briefly, or is it described in reference 9?
P2/L56-58: When introducing the course, it might be good to clarify whether it is mainly self-directed or teacher-led.
Materials and methods
The way to organise and structure participants’ characteristics vary across journals. As a reader, I prefer to read about the sample characteristics (of both participants and facilitators) in the methods section.
P4/L151: It would also be beneficial to provide some examples of the items, including the Death Literacy Index Questionnaire response format already in the Methods section.
P4/L160: Also, the interview guide could be described (on what topics, examples of the open-ended questions).
P4/L163: In an article full of abbreviations, it would be helpful to reduce the number of abbreviations. Could “KB” be written in its entirety? Consider writing the Highland Hospice rather than “HH” (if the text limit allows it).
I recommend adding a figure showing the study design as a timeline: what was measured before the course and what was measured after the course, and the people involved (course participants and facilitators) at the different measurement points, data collection techniques, and outcome variables. It would provide an overview of the mixed methods approach, including pre and post measures, so the reader can quickly grasp the study design.
“… the mixed method study design and presence of quantitative and qualitative data for LAT participants, and solely qualitative data for facilitators …”
“An online survey contained both quantitative and qualitative items and was distributed solely to LAT participants. The semi-structured interviews were conducted with both LAT participants and facilitators.”
“Thematic analysis of both participants’ free text survey responses and interview data …”
P4/L168: Motivate presenting descriptive statistics only (no significance testing).
Data analysis
P5/L202-210: I suggest this paragraph, including tables 1 and 2, can be moved to the Methods section.
P5: Table 1 – column headings in subtables are missing (frequencies and standard deviation(?), occupation, civic status, education etc.).
P6: Table 2. Column headings are missing (response, frequencies, percentage …)
P6/L216-219: I suggest this paragraph can be moved to the Methods section.
Overall Impact of LAT on Participants
The thematic analysis gives rich insights. However, I suggest removing a couple of quotes because they do not add much (or I might have missed it. If so, consider extending the excerpt-commentary units with more clarifying comments).
P12/L339-341: This quote could be removed.
P13/L343-344: This quote could be removed.
P15/L476: How does this quote connect with the preceding comment?
Discussion
P19/L658: A missing word > “This endorses the view ….”
P19/L663: DNACPR?
Conclusions
P21/L744-760: Move “limitations” to the Discussion section. Keep the “Conclusions” short based on the findings.
Author Response
The authors would like to thank the Reviewer for teh helpful suggestions and comments.
|
Reviewer 2 |
Responses |
|
P2/L56-58: When introducing the course, it might be good to clarify whether it is mainly self-directed or teacher-led.
|
We have inserted additional explanatory text to clarify. |
|
The way to organise and structure participants’ characteristics vary across journals. As a reader, I prefer to read about the sample characteristics (of both participants and facilitators) in the methods section.
|
Moved to the methods section as suggested. |
|
P4/L151: It would also be beneficial to provide some examples of the items, including the Death Literacy Index Questionnaire response format already in the Methods section.
|
Included as Appendix 1 |
|
P4/L160: Also, the interview guide could be described (on what topics, examples of the open-ended questions).
|
Included as – LAT Facilitator Interview Topic Guide Appendix 2 LAT Participant Interview Topic Guide Appendix 2 |
|
P4/L163: In an article full of abbreviations, it would be helpful to reduce the number of abbreviations. Could “KB” be written in its entirety? Consider writing the Highland Hospice rather than “HH” (if the text limit allows it).
|
We have reduced the abbreviations as suggested by writing Highland Hospice consistently throughout the manuscript and by writing authors’ names in full when describing data analysis. |
|
I recommend adding a figure showing the study design as a timeline: what was measured before the course and what was measured after the course, and the people involved (course participants and facilitators) at the different measurement points, data collection techniques, and outcome variables. It would provide an overview of the mixed methods approach, including pre and post measures, so the reader can quickly grasp the study design. “… the mixed method study design and presence of quantitative and qualitative data for LAT participants, and solely qualitative data for facilitators …”
|
Participant responses on questionnaire items were captured using a one-time survey from two time periods - pre and post LAT. They were asked to provide data for items on the questionnaire pre and then post LAT training where they were reflecting back on their perceptions / experiences before attending LAT rather than having completed the survey at different time points.
|
|
P4/L168: Motivate presenting descriptive statistics only (no significance testing).
|
The sample size was too small for any significance testing and hence not tested. |
|
P5/L202-210: I suggest this paragraph, including tables 1 and 2, can be moved to the Methods section.
|
Moved to the methods section as suggested. |
|
P5: Table 1 – column headings in sub tables are missing (frequencies and standard deviation(?), occupation, civic status, education etc.). P6: Table 2. Column headings are missing (response, frequencies, percentage …)
|
Tables reformatted as suggested |
|
P6/L216-219: I suggest this paragraph can be moved to the Methods section.
|
Moved to the methods section as suggested. |
|
The thematic analysis gives rich insights. However, I suggest removing a couple of quotes because they do not add much (or I might have missed it. If so, consider extending the excerpt-commentary units with more clarifying comments). P12/L339-341: This quote could be removed. P13/L343-344: This quote could be removed. |
Thank you for highlighting this issue. We have revised accordingly to remove |
|
P15/L476: How does this quote connect with the preceding comment?
|
The participant here suggests that they would like to have their own script / notes – I have delivered the preceding comment to reflect this. |
|
P19/L658: A missing word > “This endorses the view ….”
|
We have restructured this sentence to aid clarity |
|
P19/L663: DNACPR?
|
We have re-written this in full the first time this is used to aid clarity |
|
P21/L744-760: Move “limitations” to the Discussion section. Keep the “Conclusions” short based on the findings.
|
We have moved the limitations as suggested.
|
Reviewer 3 Report
The article describes the evaluation of an online course to help in last aid.
The study is interesting, the objectives of the article are clear, and the conclusions are valid. Although the study was carried out with only 26 individuals, I believe that this may be enough people to validate the study.
However, it is a long article, which due to some formatting problems makes it difficult to read, but overall, the article is well written, and the language seems adequate to me.
Note that I don't know if the formatting problems are introduced by some pre-editing tool.
The summary is well written, with nothing more to point out.
The introduction provides a good framework for the topic and the section on materials and methods is thoroughly explained.
The results section, section 3, is extensive and here some formatting problems start making the reading quite complicated. Namely, in tables 1, 2, and 3 the headers in the version I received are totally dark, so they don't have any information on the values\caption columns. For example, in table 1, I noticed that the third column is the percentage, but in some cases, the decimal number is clearly missing.
In figure 1, I recommend using color in the graphics. The subsections of this section should be highlighted in some way, the typeface and dimension don´t change and it´s confusing for the reader. Also, I can't find a pattern, sometimes italics are used, sometimes not, even compared to the other sections. Other times appear in italic and list but go from (a), line 609 to (c), line 618, without (b)? A real mess.
Overall, the topic is interesting and relevant, but without improvements in section 3, it doesn't seem suitable for publication.
But I want to make it clear that the formatting problems may have been caused by some editing tool.
Author Response
The authors would like to thank the Reviewer for the helpful suggestions and comments.
|
Reviewer 3 |
Responses |
|
The results section, section 3, is extensive and here some formatting problems start making the reading quite complicated. Namely, in tables 1, 2, and 3 the headers in the version I received are totally dark, so they don't have any information on the values\caption columns. For example, in table 1, I noticed that the third column is the percentage, but in some cases, the decimal number is clearly missing.
|
Tables reformatted as required |
|
In figure 1, I recommend using color in the graphics. The subsections of this section should be highlighted in some way, the typeface and dimension don´t change and it´s confusing for the reader. Also, I can't find a pattern, sometimes italics are used, sometimes not, even compared to the other sections. Other times appear in italic and list but go from (a), line 609 to (c), line 618, without (b)? A real mess.
|
Tables reformatted as required |